# A Lagrangian Thin-Shell Finite Element Method for Interacting Particles on Fluid Membranes

**DOI:** 10.3390/membranes12100960

**Published:** 2022-09-30

**Authors:** Sanjay Dharmavaram, Xinran Wan, Luigi E. Perotti

**Affiliations:** 1Department of Mathematics, Bucknell University, 1 Dent Drive, Lewisburg, PA 17837, USA; 2Language Technologies Institute, Carnegie Mellon University, 5000 Forbes Avenue, Pittsburgh, PA 15213, USA; 3Mechanical and Aerospace Engineering Department, University of Central Florida, 12760 Pegasus Drive, Orlando, FL 32816, USA

**Keywords:** interacting particles, lipid membranes, subdivision finite element, Helfrich–Canham model, model for protein–membrane interaction

## Abstract

A recurring motif in soft matter and biophysics is modeling the mechanics of interacting particles on fluid membranes. One of the main outstanding challenges in these applications is the need to model the strong coupling between the substrate deformation and the particles’ positions as the latter freely move on the former. This work presents a thin-shell finite element formulation based on subdivision surfaces to compute equilibrium configurations of a thin fluid shell with embedded particles. We use a variational Lagrangian framework to couple the mechanics of the particles and the substrate without having to resort to ad hoc constraints to anchor the particles to the surface. Unlike established methods for such systems, the particles are allowed to move between elements of the finite element mesh. This is achieved by parametrizing the particle locations on the reference configuration. Using the Helfrich–Canham energy as a model for fluid shells, we present the finite element method’s implementation and an efficient search algorithm required to locate particles on the reference mesh. Several analyses with varying numbers of particles are finally presented reproducing symmetries observed in the classic Thomson problem and showcasing the coupling between interacting particles and deformable membranes.

## 1. Introduction

A two-dimensional interacting assembly of particles embedded, yet free to move, on a deformable fluid membrane is a recurring motif in material science, soft-matter, and biophysical systems. This type of assembly serves as a model to understand diverse phenomena, such as protein packing on viral capsids [1], cellular processes such as endo-/exo-cytosis [2], and crystallization on curved geometries [3,4,5,6]. In these applications, the two-dimensional (2D) nature of the structure is often critical to the application function. Ever since the discovery of graphene, it has been well known that 2D materials have unique electronic, mechanical, and optical properties, which distinguish them from their bulk 3D counterparts [7]. Consequently, in recent decades there has been significant interest in synthesizing and characterizing novel 2D materials with tailorable properties. Colloidal particles, owing to their versatility and tunability, are playing a significant role in this regard [8,9,10].

In addition to the practical applications noted above, 2D particle assemblies on thin substrates are also adopted to probe fundamental questions on the structure of matter. Numerous, particle-based experimental studies have focused on understanding crystal structures and defects on curved geometries [11,12,13,14,15,16,17]. Curvature and geometrical frustration induce crystalline states not found in flat geometries. For instance, while on flat infinite substrates, particles under isotropic repulsive interactions arrange themselves in a regular hexagonal lattice, on curved substrates (especially for a sufficiently large number of particles) the arrangements invariably contain crystal defects [11,18].

Despite growing interest and advances in numerical methods to model colloids on two-dimensional surfaces, most studies have focused on rigid geometries, i.e., the substrate cannot deform in response to the forces exerted by the particles. Some studies do consider the adhesion effects of colloidal particles on the shape of the substrate [19,20], but remain focused on the *local* distortion effects. *Long-range* and *cooperative* interactions between particles and substrates have largely not been explored.

A significant challenge in studying such particle systems using computational models is the coupled interaction between the particles and the deformable substrate. Since the particles are constrained to move on the substrate whose shape is itself determined by particle interactions, the configuration of the particles and the substrate’s shape cannot be independently determined. This poses unique computational challenges. Current approaches impose constraints to anchor the particles on the substrate [21] or artificially restrict the particles to lie at the nodes of the mesh used to discretize the substrate [22,23]. Relying on constraints to force particles on the substrate is computationally expensive and cumbersome to implement, while restricting the particles’ positions to the nodes of the substrate mesh imposes undue restrictions on the allowed particle/substrate equilibrium configurations. In a recent work [24], we developed an alternative method that is efficient and does not introduce approximations or spurious constraints in modeling the particles–substrate interactions. In this *Lagrangian* approach, we parametrize the location of the particles using coordinates mapped on the *reference* configuration. Together with the deformation mapping of the substrate, these reference coordinates constitute the system’s degrees-of-freedom. To obtain the actual particles’ locations on the deformed substrate, we compose the reference coordinates with the substrate’s deformation map. Thus, constraints are not required to anchor the particles to the surface. We applied this framework to study interacting particles on a spherical fluid shell, employing a spectral Galerkin method and accordingly discretizing the surface deformation map using a spherical harmonic expansion. To circumvent computational issues stemming from the in-plane fluidity of the substrate [25,26,27], a *radial graph* ansatz [28] was implemented. According to this ansatz, the displacement is parametrized along the radial direction from the center of the reference sphere. This is similar to the Monge representation used for flat membranes [29,30,31].

Although novel and effective in modeling the particle–substrate coupled interactions correctly, this approach presented two key limitations. The radial graph ansatz implies that only radial displacements from the spherical reference state can be computed. This restricts the equilibrium configurations that can be obtained, precluding severely distorted shapes as the ones appearing during endo-/exo-cytosis of cells and budding of viral capsids. An example of such a state in multi-phase lipid vesicles is shown in Figure 2h of [32]. The second major limitation regards the use of the spectral Galerkin method. As spherical harmonics have to be computed using recursive algorithms, this approach is computationally expensive, especially when high order terms are required to better represent the deformed configuration. In addition, the number of spherical harmonics needed up to order *ℓ* grows quadratically (i.e., ${\left(\ell +1\right)}^{2}$), leading to a rapid rise in computational cost as higher accuracy is sought. The global nature of these functions and the coordinate singularities at the sphere’s poles also requires very high-order quadrature rules. Finally, due to Gibbs-type oscillations, the method is inadequate when equilibrium configurations present sharp shape changes.

Even though our studies have so far focused on “zero temperature” equilibrium states, where thermal fluctuations do not play a role, the ultimate application of the computational models lies in finite-temperature studies, often simulated using Monte Carlo methods. Indeed, for many biophysical and soft-matter systems at physiologically relevant temperatures, thermal fluctuations play an important role. However, applying the spectral Galerkin method in Monte Carlo simulations is not straightforward. Monte Carlo simulations typically perturb the system’s degrees-of-freedom with “moves” that must satisfy the *detailed balance* condition. In spectral methods, the degrees-of-freedom are the mode coefficients that do not have a clear physical interpretation and enforcing *detailed balance* is not straightforward. A finite element approach with nodal displacement degrees of freedom, on the other hand, is amenable to standard Monte Carlo approaches, and can, therefore, be easily adapted to simulate finite-temperature systems.

In this article, we extend our previous work and remove the limitations described above. We adapt the Lagrangian particle formulation to the context of $C^{1}$ conforming thin shell finite elements [33] that use interpolation functions based on a Loop subdivision scheme [34,35]. The Loop subdivision finite element scheme requires only displacement degrees of freedom while retaining $C^{1}$ continuity of the shape functions across elements. This is necessary to compute the curvatures terms appearing in the thin-shell elastic energy and the mapping of the particles from the reference to the deformed mesh (as described in Section 2.2 and Section 2.3, this step may require the evaluation of the deformation gradient tensor at the interface between elements). Furthermore, this finite element method (FEM) is widely used in lipid membrane studies [25,26,27] and including the Lagrangian particle approach into this framework can help other researchers in the area.

In addition to merging the Lagrangian particle formulation with thin-shell FEM and removing the radial graph ansatz, an essential contribution of the present work are the details of the search algorithm used to locate the particles on the reference mesh. Since, in our Lagrangian approach, the particle positions are parametrized in the reference configuration, the search for the elements to which each particle belongs is carried out only on the reference mesh.

This paper is organized as follows. In Section 2.1, we present the variational formulation of the substrate-particles system. We model the fluid substrate using the Helfrich–Canham energy and the particle interactions using pair-wise isotropic potentials. The in-plane fluidity of the substrate poses computational challenges, which are addressed using the *gauge-fixing procedure* proposed in [27]. In Section 2.2, we present the Lagrangian formulation for the gauge-fixed formulation of the Helfrich–Canham energy. We present the weak form of the equilibrium equations which are then used to discretize the system. We briefly comment on the implementation aspects of the Loop subdivision FEM method relevant to the present problem in Section 2.3. In Section 2.4, the details for the proposed search algorithm are provided. Finally, in Section 3, we discuss the details of a validation study and the results of representative analyses to showcase the presented method.

## 2. Materials and Methods

This work considers a system consisting of a topologically spherical fluid shell with *N* mutually interacting particles. The particles are embedded in the substrate but can freely move along its surface. We focus on equilibrium configurations and assume that the particles do not experience in-plane viscous forces. Since the particles cannot leave the surface, the substrate can exert only normal forces on the particles and vice versa. This model system (Figure 1) is motivated, for example, by the study of interacting proteins embedded on lipid membranes. Note that, following studies in the computational biophysics literature, in this work the term “membrane” is often used to refer to a fluid shell, despite the difference in connotations of the two terms in continuum mechanics.

### 2.1. Model Formulation

Without loss of generality, we assume that the substrate’s undeformed configuration is a unit sphere ($S^{2}$). Let us denote the surface of the deformed configuration as $\omega \subset R^{3}$ and $f:S^{2}\to \omega$ as the deformation map of the deformed surface. We model the bending elastic energy $E_{HC}$ of the substrate using the Helfrich–Canham energy [36]:(1)$$E_{HC}=\int_{\omega }\kappa {\left(H-C_{0}\right)}^{2}+\kappa_{g}K\;da\;,$$
where *H* and *K* are the mean and Gaussian curvatures of ω with κ and $\kappa_{g}$ their associated bending stiffnesses, and $C_{0}$ is the preferred curvature. Helfrich–Canham energy is a widely used model for lipid membranes, and the energy (Equation 1) is expressed as an integral on the current configuration ω with da being its area measure. This is in line with the observation that a lipid membrane behaves like a two-dimensional fluid, and, consequently, has no preferred in-plane reference configuration. Since the stretching modulus of a lipid membrane is significantly larger than the bending moduli, the membrane is usually approximated as area-preserving [37]. This incompressibility condition is enforced through the area constraint:(2)$$\int_{\omega }\;da=4\pi \;.$$

We model particle interactions with a pair potential Φ(r), where *r* represents the 3D Euclidean distance between the interacting particles. The total interaction energy is thus given by:(3)$$U\left(x_{1},\cdots ,x_{N}\right)=\sum_{i=1}^{N}\sum_{j>i}^{N}\Phi (||x_{i}-x_{j}||)\;,$$
where $x_{i}$ represents the 3D position of particle *i* on the membrane and ||·|| denotes the standard Euclidean norm. In this work, we will explore various potentials, including Coulombic, harmonic, and Lennard–Jones interaction potentials (see Section 3).

The (Helmholtz) free energy E of the system is given by:(4)$$E=E_{HC}+U\left(x_{1},\cdots ,x_{N}\right)-pV\;,$$
where *p* is the internal osmotic pressure on the substrate and *V* is the enclosed volume. The area constraint in Equation (Equation 2) must be additionally imposed. In the simulations presented in this work, we do so using a penalty method. Alternatively, the area constraint can be enforced using a Lagrange multiplier, which can be numerically implemented using the Augmented Lagrangian method.

**Remark** **1.**
*Instead of a fixed pressure applied on the membrane, it is common to constrain the volume enclosed in its interior [25,26]. Under such a formulation, we define a non-dimensional parameter $v:=V/V_{0}$ that measures the fraction of volume enclosed by ω with respect to the volume enclosed by the unit (reference) sphere ($V_{0}=4\pi /3$). Thus, we impose the condition:*

(5)
$$\frac{1}{3}\int_{\omega }f\cdot d\;da=\frac{4\pi v}{3}\;,$$

*where d is the outward pointing (unit) normal to surface ω and f is the deformation map. The integral on the left side computes the total volume enclosed by ω, and has been obtained using the divergence theorem: $\int_{V}\mathrm{div}\;f\;dv=\int_{\omega }f\cdot d\;da$, where divf=3. When required, this volume constrained formulation is employed in the simulations presented in Section 3.*


#### 2.1.1. In-Plane Fluidity

In-plane fluidity, also known as *reparametrization invariance*, is a defining material symmetry for fluid shells. For instance, in lipid membranes it arises because lipid molecules can freely move on the surface facing very little resistance, conferring to the membrane a 2D fluid-like behavior. In the Helfrich–Canham model, fluidity manifests as reparametrization invariance of the energy shown in Equation (Equation 1). This can be inferred from the dependence of the energy solely on differential geometrical quantities, which are independent of the chosen parameterization of the surface. Thus, unlike the case of solid shells, in-plane fluidity of fluid membranes implies that the latter does not have a prescribed “reference configuration”.

A consequence of in-plane fluidity is that the tangential components of the Euler-Lagrange equations vanish identically [37,38]. Only the normal components of the Euler–Lagrange equations contribute to the equilibrium equations and, thus, the equations are inherently underdetermined. It can be shown that equilibrium configurations, when they exist, belong to an *equivalence class* of solutions [27]. Members of this class are different parametric representations for the *same* surface and are related to each other by a diffeomorphism map between the parameter spaces. Since there are infinitely many diffeomorphisms between spherical surfaces, they give rise to a grossly redundant solution space. This causes significant challenges in numerical simulations where spurious zero-energy shear modes [26] and severe mesh distortions [25] are commonly reported.

**Remark** **2.**
*Note that, although the system considered here includes particles in addition to a fluid substrate, the analysis presented above still applies. In-plane fluidity remains the material symmetry for the problem. This is because the particles are free to move on the surface of the membrane and they lack preferred reference positions. This is evidenced by Equation (Equation 3), where the particles’ interaction energy only depends on the current particles’ locations, i.e., $\left\{x_{i}\right\}$. Thus, reparametrizing the surface will neither change the positions of the particles nor their interaction energy. Therefore, U is reparametrization invariant, and so is E in Equation (Equation 4). As the numerical issues noted in the previous paragraph will also plague the membrane–particle system, in the next section we describe a strategy to circumvent these issues.*


#### 2.1.2. Gauge-Fixing Procedure

Several numerical schemes have been developed to address computational issues stemming from reparametrization invariance. Some methods impose local area incompressibility (instead of global area constraint) [26] to suppress zero-energy modes, while others [25,39] dampen tangential motion by introducing ad hoc in-plane energies whose contributions are iteratively decreased to zero. Monge representation [29] or a *radial graph* ansatz [28,40]—where the surface is parametrized using a single unknown function—have also been used to circumvent reparametrization invariance. In this work, we employ the *gauge-fixing procedure* recently proposed in [27], which is computationally efficient for topologically spherical surfaces and does not require any iterative reduction in ad hoc energy terms that change the physics of the model. In this approach, reparametrization invariance is viewed as a form of *gauge symmetry*, a symmetry of a physical theory whose energy/action functional is invariant under certain continuous group of transformations. Gauge theories always contain redundant degrees of freedom and have underdetermined Euler–Lagrange equations. Additional constraints ought to be imposed to break this redundancy in what constitutes the *gauge-fixing procedure*. We shall now summarize this procedure for spherical fluid membranes as presented in [27].

The gauge-fixing procedure entails supplementing the free energy E (cf., Equation (Equation 4)) with an additional term, i.e.,:(6)$$\tilde{E}=E+E_{\mathrm{HM}}\;,$$
where the second term is the *harmonic map energy* given by:(7)$$E_{\mathrm{HM}}=\int_{\omega }\frac{1}{2}g^{\alpha \beta }h_{\alpha \beta }\;da\;.$$

Here, $g_{\alpha \beta }$ (α,β=1,2) are the (contravariant) components of the metric tensor of the deformed surface ω, and $h_{\alpha \beta }$ are the (covariant) components of the metric tensor of the reference surface $S^{2}$. Although it might seem that by adding $E_{\mathrm{HM}}$ (Equation (Equation 7)) to the free energy, we are modifying the surface constitutive law, this is not the case. Despite the additional harmonic map term in Equation (Equation 6), at an equilibrium state, $E_{\mathrm{HM}}$ does not alter the Euler-Lagrange equations of the Helfrich–Canham energy. To see this, let us first note that the tangential components of the Euler-Lagrange equation due to E are trivially zero [37,41]. Thus, the only contribution to the Euler–Lagrange equations of (Equation 6) in the tangential direction is due to $E_{\mathrm{HM}}$. The tangential variation of $E_{HM}$ is explicitly given by [27]:(8)$$\delta_{\left|\right|}E_{\mathrm{HM}}=\int_{\omega }h_{\alpha \beta }\left[\frac{1}{\sqrt{g}}\partial_{\mu }\left(\sqrt{g}g^{\alpha \mu }\right)+g^{\mu \gamma }\Upsilon_{\mu \gamma }^{\alpha }\right]v^{\beta }\;da\;,$$
where $\Upsilon_{\cdot \cdot }^{\cdot }$ is the Christoffel symbol corresponding to the metric tensor *h* and $v^{\beta }$ is the tangential variation along ω. The term in the brackets in Equation (Equation 8) (for each β∈{1,2}) is the tangential component of the Euler-Lagrange equations of the harmonic map. The normal component is given by:(9)$$\delta_{⊥}E_{\mathrm{HM}}=\frac{1}{2}\int_{\omega }h_{\alpha \beta }\left[2b^{\alpha \beta }-2Hg^{\alpha \beta }\right]w\;da\;,$$
where *w* is a smooth variation normal to ω.

It has been shown in Theorem 3, Appendix D of [27] that

**Theorem** **1.**
*If*

(10)
$$\frac{1}{\sqrt{g}}\partial_{\mu }\left(\sqrt{g}g^{\alpha \mu }\right)+g^{\mu \gamma }\Upsilon_{\mu \gamma }^{\alpha }=0,\;\;\mathrm{for}\;\alpha \in \left\{1,2\right\},$$

*then $\delta_{⊥}E_{\mathrm{HM}}\equiv 0$ for all smooth variations w.*


That is, if an equilibrium state satisfies the tangential component of the Euler–Lagrange equations due to $E_{\mathrm{HM}}$, then the contribution to the normal component of the Euler–Lagrange equations due to $E_{\mathrm{HM}}$ is trivially zero. Thus, the normal component of the equilibrium equation that determines the lipid membrane’s shape remains unmodified. In other words, at equilibrium, the harmonic map energy only contributes to the Euler–Lagrange equations’ tangential components and, in doing so, provides the constraints necessary to prevent arbitrary tangential motions. The equilibria for the gauge-fixed and the original formulation are identical.

The principal advantage of the gauge-fixing procedure is that adding the harmonic map energy $E_{\mathrm{HM}}$ in Equation (Equation 6) does not change the physics of the problem. Furthermore, the gauge-fixed formulation is computationally more efficient than the previously mentioned iterative formulations, since no additional energy terms are included that need to iteratively be reduced to zero.

The gauge-fixing procedure developed in [27] only includes $E_{\mathrm{HC}}$ while E (c.f., Equation (Equation 4)) also includes the particle interaction energy, *U*. However, this inclusion does not affect the gauge-fixing procedure as the only relevant fact is that the energy is reparametrization invariant. In light of Remark  2, we note that since the particles are also “fluid-like”, $\tilde{E}$ is also reparametrization invariant and the gauge-fixing procedure can be applied to the present problem.

One notable feature of Equation (Equation 6) is that unlike E (which was invariant under arbitrary reparametrizations of the surface), the *gauge-fixed* formulation, i.e., $\tilde{E}$, is invariant under conformal reparametrizations of the surface. Thus, the gauge-fixing procedure does not break completely the reparametrization invariance symmetry. It has been shown in [27] that configurations that remain in the equivalence class are related by the six-dimensional Möbius group of transformations. These six modes can be viewed as the rigid translation and rotation modes of the sphere [42]. The three translational modes can be constrained by imposing the “zero-mass” constraint [27]:(11)$$\int_{\omega }R\left(X^{1},X^{2}\right)\;da=0\;,$$
where $R(X^{1},X^{2})$ is a parametrization of the reference sphere. The three rotational modes can, in theory, be fixed by landmark constraints [43]. In practice, however, we found that reliable results (presented in Section 3) could be obtained without any landmark constraints when we used the L-BFGS [44] numerical minimization algorithm.

### 2.2. Lagrangian Particle Formulation

One of the challenges in discretizing $\tilde{E}$ is that the particles must always lie on the surface ω that is itself determined by the configuration of the particles. The shape of the surface and the arrangement of the particles are inextricably coupled, and one cannot be solved independently of the other. In [24], we have developed a variational *Lagrangian* formulation to address this challenge. In this formulation, the particles’ 3D positions on ω are parametrized by the particles’ coordinates $X_{i}$ on the reference surface $S^{2}$, which, therefore, become the particles’ degrees of freedom. Since the particles are fluid-like and do not have a preferred reference configuration, $X_{i}$ have no physical significance. To obtain the actual 3D position $x_{i}$ of the particles, we compose $X_{i}$ with the deformation map $f:S^{2}\to \omega$, i.e.,  $x_{i}=f\left(X_{i}\right)$. This is illustrated schematically in Figure 2. Here, $x_{i}$ is the position of particle *i* on ω, the relevant variable that appears in the interaction energy *U* (see Equation (Equation 3)). Using the Lagrangian formulation, we can write the free energy in Equation (Equation 4) with respect to the reference surface as:(12)$$E\left[X_{1},\cdots ,X_{n},f\right]=\int_{S^{2}}\left[\kappa {\left(H-C_{0}\right)}^{2}+\kappa_{g}K\right]J\;dA-pV+U\left(f_{1},f_{2},\cdots ,f_{N}\right)\;,$$
where $J=\sqrt{g}/\sqrt{G}$ is the Jacobian determinant, expressed in terms of *g* and *G*, the determinants of the metric tensor of ω and $S^{2}$, respectively, and we adopted the notation $f_{i}:=f\left(X_{i}\right)$. The area measure on $S^{2}$ is denoted as dA. The degrees-of-freedom for the system are $X_{1},\cdots X_{N}$, and f, which are explicitly indicated in (Equation 12).

Equilibrium configurations are determined by finding critical points of (Equation 12) (or, the gauge-fixed formulation (Equation 6)). In a variational setting, this requires taking the variations of (Equation 12) with respect to the degrees-of-freedom, viz., f and $X_{i}$ (i=1,⋯N). Let η(X) denote the variation in f(X) and $\theta_{i}$ the variation in $X_{i}$. As discussed in [24], care must be taken while computing the variation in $f_{i}=f\left(X_{i}\right)$. For each i∈{1,2,⋯,N} we obtain:(13)$$\delta f_{i}=\delta \left[f\left(X_{i}\right)\right]=\eta \left(X_{i}\right)+\nabla f\left(X_{i}\right)\theta_{i}\;,$$
where ∇f is the deformation gradient tensor expressed in terms of the surface gradient ∇(·). Using basis vectors $a_{\alpha }:=\partial f/\partial X^{\alpha }$ (α=1,2) for the tangent space to ω at X, we can write $\nabla f\left(X\right)=g^{\alpha \beta }a_{\alpha }⊗a_{\beta }$, where $g^{\alpha \beta }$ are the contravariant components of the metric tensor (defined explicitly in the next section). We see from (Equation 13) that $\delta f_{i}$ has two contributions: the first one accounts for variation in the position of the substrate, and the second one accounts for the variation of the particle positions in the reference configuration $\theta_{i}$, which are pushed forward to the current configuration by the deformation gradient tensor. Note that all the tensor and vector fields in (Equation 13) are evaluated at $X_{i}$.

#### Weak Form

In this section, we derive the weak-form for the gauge-fixed formulation, cf. (Equation 6), (Equation 7), and (Equation 12). For clarity, we shall begin by recalling some notation and important differential geometry identities that we will use to derive the weak form. Much of what we present in this section, especially concerning the weak form for the gauge-fixed Helfrich–Canham energy, can be found in [27]. The computation of the first variation of the particles-substrate energies using the Lagrangian formulation discussed in the previous section is based on our previous work [24].

The basis for the tangent space of ω at X (parametrized by $X^{1}$ and $X^{2}$) is spanned by
(14)$$a_{\alpha }:=f_{,\alpha }=\frac{\partial f}{\partial X^{\alpha }}\in R^{3},\;\alpha \in \{1,2\},$$
where we use ${\left(\cdot \right)}_{,\alpha }=\partial \left(\cdot \right)/\partial X^{\alpha }$. The components of the metric tensor of *ω* are given in terms of the Euclidean dot product in
$R^{3}$ by
$$g_{\alpha \beta }=a_{\alpha }\cdot a_{\beta }\;.$$


The dual basis vectors of $a_{\alpha }$ are denoted as
$$a^{\alpha }:=g^{\alpha \beta }a_{\beta }\;,$$
where $g^{\alpha \beta }$ are the contravariant components of the metric tensor and we used the Einstein summation convention on the repeated indices. This convention is employed in the following discussion as well. The unit normal field to ω is given by
$$d=\frac{a_{1}\times a_{2}}{\sqrt{g}}\;,$$
where we have used the identity $\sqrt{g}=\left|\right|a_{1}\times a_{2}\left|\right|$.

The components of the second fundamental form of ω are given by
$$b_{\alpha \beta }=-d_{,\alpha }\cdot a_{\beta }=d\cdot a_{\alpha ,\beta }\;.$$

The mean and Gaussian curvatures can then be computed as, respectively:(15)$$H=\frac{1}{2}b_{\alpha }^{\alpha }=-\frac{1}{2}a^{\alpha }\cdot d_{,\alpha }\;,\;K=\det\left(b_{\beta }^{\alpha }\right)\;,$$
where, as per convention, tensor indices are lowered or raised by multiplying with $g_{\alpha \beta }$ or its inverse $g^{\alpha \beta }$.

To derive the weak form of the gauge-fixed functional, $\tilde{E}$, we compute its variation $\delta \tilde{E}$ with respect to η (the variation in f) and $\theta_{i}$ (the variation in $X_{i}$). It follows from Equation (Equation 6) that
(16)$$\delta \tilde{E}=\delta E+\delta E_{\mathrm{HM}}\;.$$

The first variation on the right side is computed by taking the variations of all the contributing terms in Equation (Equation 4). To do so, let us first define the following quantities:(17)$$\begin{array}{c} n^{\alpha }=\left[\kappa \left(H-C_{0}\right)+2\kappa_{g}H\right]g^{\alpha \beta }d_{,\beta }+\left[\kappa {\left(H-C_{0}\right)}^{2}+\kappa_{g}\left(K-b_{\beta }^{\nu }a^{\beta }\cdot d_{,\nu }\right)\right]a^{\alpha }+\\ \gamma \sqrt{a}a^{\alpha }-\frac{1}{3}pV\left[\left(f\cdot d\right)a^{\alpha }-\left(f\cdot a^{\alpha }\right)d\right]\;, \end{array}$$
(18)$$m^{\alpha }=\left[-\kappa \left(H-C_{0}\right)\delta_{\beta }^{\alpha }+\kappa_{g}\left(b_{\beta }^{\alpha }-2H\delta_{\beta }^{\alpha }\right)\right]a^{\beta }\;,$$
where $\delta_{\beta }^{\alpha }$ is the Kronecker delta and γ is the Lagrange multiplier enforcing the area constraint (see Equation (Equation 2)). It can be shown (see [25,27] for details) that:(19)$$\begin{array}{c} \delta E=\int_{S^{2}}\left[n^{\alpha }\cdot \delta a_{\alpha }+m^{\alpha }\cdot \delta d_{,\alpha }-\frac{1}{3}pVd\cdot \eta \right]\;JdA+\\ \sum_{i=1}^{N}\sum_{j>i}^{N}\Phi^{\prime }\left(r_{ij}\right)\frac{\left(f_{i}-f_{j}\right)}{\left|\right|f_{i}-f_{j}\left|\right|}\cdot \left(\delta f_{i}-\delta f_{j}\right)\;, \end{array}$$
where $r_{ij}=\left|\right|f_{i}-f_{j}\left|\right|$, and the variations $\delta a_{\alpha }$ and $\delta d_{,\alpha }$ are given by [26]:(20)$$\delta a_{\alpha }=\eta_{,\alpha }\;,$$
(21)$$\delta d_{,\alpha }=-\left[a_{,\alpha }^{\beta }⊗d+a^{\beta }⊗d_{,\alpha }\right]\cdot \eta_{,\beta }-\left[a^{\beta }⊗d\right]\cdot \eta_{,\beta \alpha }\;,$$
where ⊗ is the tensor product of vectors in $R^{3}$. The last term in Equation (Equation 19) was obtained by taking the variation of *U*, cf. Equation (Equation 3), using $x_{i}=f\left(X_{i}\right)$. Equation (Equation 19) can be further simplified using Equation (Equation 13) to obtain:(22)$$\begin{array}{c} \delta E=\int_{S^{2}}\left[n^{\alpha }\cdot \delta a_{\alpha }+m^{\alpha }\cdot \delta d_{,\alpha }-\frac{1}{3}pvd\cdot \eta \right]\;JdA+\\ \sum_{i=1}^{N}\sum_{j>i}^{N}\left[\left[g_{ij}\cdot \eta \left(X_{i}\right)-g_{ij}\cdot \eta \left(X_{j}\right)\right]+\left[g_{ij}\cdot a_{\alpha }\left(X_{i}\right)\theta_{i}^{\alpha }-g_{ij}\cdot a_{\alpha }\left(X_{j}\right)\theta_{j}^{\alpha }\right]\right]\;, \end{array}$$
where $\theta_{i}^{\alpha }$ (α=1,2) are the components of θ, i.e., $\theta_{i}:=\theta_{i}^{\alpha }a_{\alpha }$, and 
(23)$$g_{ij}:=\Phi^{\prime }\left(r_{ij}\right)\frac{\left(f_{i}-f_{j}\right)}{\left|\right|f_{i}-f_{j}\left|\right|}\;.$$

The second term on the right side of Equation (Equation 16) can be evaluated by taking the variation of Equation (Equation 7), leading to [27],
(24)$$\delta E_{\mathrm{HM}}=\frac{1}{2}\int_{S^{2}}h_{\alpha \beta }\left(-2g^{\alpha \gamma }a^{\beta }+g^{\alpha \beta }a^{\gamma }\right)\cdot \eta_{,\gamma }J\;dA\;.$$

Finally, the weak form of the equilibrium equations is given by the condition:(25)$$\delta E+\delta E_{\mathrm{HM}}=0\;,$$
for all admissible variations $\eta :S^{2}\to R$ and $\theta_{i}^{\alpha }\in R$, (i=1,2,⋯,N and α=1,2). If requested, the area constraint (Equation (Equation 2)) must also be imposed.

We conclude this section with a remark. A minimization solver cannot be employed to compute the equilibria of $\tilde{E}$ using the weak form in Equation (Equation 25). Indeed, since $E_{\mathrm{HM}}$ does not need to be positive, a minimizer of $\tilde{E}$ is not necessarily the minimizer of E. That is, the gauge-fixing procedure of adding $E_{\mathrm{HM}}$ to the energy could potentially turn local minima into saddle points and local maxima. To circumvent this problem, in the numerical results presented in Section 3, we minimize instead:(26)$$\hat{E}:=E+\lambda_{g}{\left(E_{\mathrm{HM}}\right)}^{2}+\mu_{1}{\left(\int_{S^{2}}J\;dA-4\pi \right)}^{2}+\mu_{2}|\int_{S^{2}}R\left(X^{1},X^{2}\right)J\;dA{|}^{2}\;,$$
where the harmonic map energy appears as the second (squared) term. The coefficient $\lambda_{g}$ is a parameter that controls the strength of the harmonic map energy. The last two terms are the area (Equation (Equation 2)) and “zero-mass” (Equation (Equation 11)) constraints, which are imposed using a penalty formulation with parameters $\mu_{1}$ and $\mu_{2}$. The weak form associated with this modified functional is given by:(27)$$\delta E+\left(2\lambda_{g}E_{\mathrm{HM}}\right)\delta E_{\mathrm{HM}}+\lambda_{1}\int_{S^{2}}a^{\alpha }\cdot \eta_{\alpha }J\;dA+\lambda_{2}\int_{S^{2}}R\left(X^{1},X^{2}\right)a^{\alpha }\cdot \eta_{\alpha }J\;dA=0\;,$$
where $\lambda_{1}=2\mu_{1}\left(\int_{S^{2}}J\;dA-4\pi \right)$ and $\lambda_{2}=2\mu_{2}\int_{S^{2}}RJ\;dA$. The last two terms in Equation (Equation 27) were obtained by taking the variations of the last two terms in Equation (Equation 26) and using the identity for variation in *J*, viz., $\delta J=Ja^{\alpha }\cdot \eta_{\alpha }$. The terms δE and $\delta E_{\mathrm{HM}}$ appearing in Equation (Equation 27) are given by Equation (Equation 22) and Equation (Equation 24), respectively. If instead of a fixed pressure *p*, a volume constraint (Equation (Equation 5)) was to be imposed, then the corresponding penalty term must be added to Equations (Equation 26) and (Equation 27).

### 2.3. Loop Subdivision Finite Element Method for Thin Shells

To discretize Equation (Equation 26) and its weak form Equation (Equation 27), we employ a Ritz–Galerkin approach. Due to the dependence of $E_{HC}$ on the mean curvature *H*, this discretization step requires the use of shape functions that ensure $C^{1}$ continuity across the elements. Furthermore $C^{1}$ continuity across the elements is also required due to Lagrangian particle formulation adopted here. Indeed, mapping particles, described in terms of their reference positions, to the deformed mesh (where the interaction energies must be calculated) requires the evaluation of the deformation gradient tensor, potentially also at the interface between elements. To satisfy the $C^{1}$ continuity requirement, we employ the thin-shell finite element method based on (Loop) subdivision surfaces first proposed in [33] for solid shells. This approach achieves $C^{1}$ continuity solely using the nodal displacement degrees-of-freedom of a triangular mesh. The shape functions are box-splines defined in terms of *barycentric coordinates* on an element. Unlike traditional finite elements, these functions are supported on a ring of neighbors around a given element (e.g., see Figure 2 in [26]). Despite having non-local support, this method is very efficient and it has been used to solve a variety of shell problems, including problems on fluid shells [25,26,27]. Explicit details of this method, including its implementation, can be found, for example, in [26,33].

A quirky feature of the method is that the shape functions are explicitly defined only in a triangular element belonging to a *regular patch*, i.e., a (triangular) element whose vertices have all valence equal to six. Elements that are not regular are called *irregular*. Evaluating shape functions at a point inside an irregular element is not straightforward. The element must be subdivided (using the Loop scheme [34]) as many times as required so that the point of interest (e.g., the location of a quadrature point or of a particle) is inside a regular (subdivision) element. Repeated subdivisions may be performed by explicitly multiplying nodal coordinates with appropriate matrix operators. However, this subdivision step can be computationally cumbersome depending on the number of required subdivisions. In [35], an elegant and efficient implementation has been proposed to evaluate shape functions on irregular elements without any explicit matrix operator multiplications. Sets of functions called *eigenbasis* required for these calculations have been tabulated in [45] for a variety of nodal valences.

The distinction between regular versus irregular elements is particularly relevant to our problem for two reasons. Firstly, since we consider topologically spherical surfaces, it is well known that it is impossible to triangulate such a surface only with regular elements. According to Euler’s theorem, at least 12 nodes with valence five are needed to cover topologically spherical surfaces. Secondly, while computing a particle’s 3D coordinates, it is necessary to evaluate the shape functions at arbitrary points on the surface. Some of these points could potentially be inside an irregular element or at the edges of one. Explicit subdivision by multiplying nodal degrees-of-freedom by the subdivision matrix (see Equation (70) in [33]) would be inefficient as this step must be performed for every particle on an irregular element and at each step of the minimization iteration. In our implementation, we therefore employ the eigenbasis discussed in [35,45] to evaluate shape functions efficiently.

### 2.4. Search Algorithm

Computing the discretized energy and the weak form requires to evaluate the shape functions at the coordinates of the particles. For example, this can be seen in Equation (Equation 22), where the variation η and function $a_{\alpha }$ appearing inside the summations must be evaluated at $X_{i}$ (i=1⋯,N). Since the shape functions are defined piece-wise over each element, an important step in the finite element implementation consists in identifying the element to which a given particle belongs. Mapping the particles to their corresponding element will necessarily involve searching over the mesh elements. This section will present an efficient search algorithm for this purpose.

We distinguish between two types of coordinates to parametrize particle positions—*global coordinates* and *local coordinates*. Global coordinates are defined continuously on the entire reference surface. In the formulation presented in Section 2.2, $X_{i}$ are the particles’ global coordinates. Since the reference surface is spherical, we can use spherical coordinates for the global coordinates, i.e., $X_{i}=\left(\vartheta_{i},\phi_{i}\right)$, where $\vartheta_{i}$ is the co-latitude angle of particle *i* measured from the positive z-axis and $\phi_{i}$ is its azimuthal angle. Local coordinates, on the other hand, are defined per element. For the Loop subdivision finite element scheme, we use barycentric coordinates (u,v,w) (with u+v+w=1) as local coordinates to locate particles on a given triangular element.

The goal of the search algorithm is to map the global coordinates of a given particle to its local (barycentric) coordinates on an element. The inputs to the algorithm are the mesh (nodes and connectivity information) and global coordinates of a given particle. The algorithm’s outputs are the local coordinates and the index of the triangular element. The local coordinate and the index will then be used to evaluate the shape functions needed in computing E and ∂E (Equations (Equation 12) and (Equation 22)).

Note that if we determine the (index of the) element containing the particle, then we can determine the local coordinates using the ray-triangle intersection algorithm (see Appendix A). Thus, the critical step in the search algorithm is to determine the element containing a given particle. This can be naively achieved using a brute-force approach: loop over all elements for each particle and determine if an element contains the particle using the ray-triangle intersection method. However, this approach is inefficient as it involves many failed and unnecessary searches.

Instead, a computationally efficient approach would be to bin the elements into subgroups depending on their relative proximity in 3D (i.e., construct a hash table). Then, given a particle position, we can use its coordinates to quickly identify the subgroup it belongs to and only search in the elements belonging to that subgroup using the ray-triangle intersection algorithm. For all the elements in the subgroup to which the particle does not belong, the ray-triangle intersection algorithm will return barycentric coordinates outside the expected [0,1] interval.

To construct the hash table, we divide $R^{3}$ into $M_{X}\times M_{Y}\times M_{Z}$ rectangular boxes, in the *X*, *Y*, and *Z* direction, respectively. We begin by scaling the mesh such that it spans the boxes. The scale factors must be stored for use later in the ray-triangle intersection step. Although optional, this step allows to use integer arithmetic in the next steps, therefore improving efficiency. We index each box with a tuple of integers (p,q,r), where *p*, *q*, and *r* are the minimum *X*, *Y*, and *Z* coordinates of points in the box. We then associate to each box the list of triangular elements that intersect the box. This is done using two loops as illustrated in Algorithm 1. We first loop over all the elements in the mesh and determine all the boxes a given element intersects. This list is called faceList in Algorithm 1. Next, we invert faceList to associate each box with all the elements contained (even partially) in that box. Note that an empty box (i.e., not containing any face) is not listed in the final hash table. The resulting hash table that associates a given box with all the faces it intersects is stored in boxList.
**Algorithm 1** Hash table algorithm.**Input: **ver← vertices, con← connectivity, $M_{X}$, $M_{Y}$, $M_{Z}\leftarrow$ Integers**Output: **A hash table with entries {box:face1,face2,…} for each box.    scale ver s.t. the mesh spans $M_{X}M_{Y}M_{Z}$ boxes    **for**face in con **do**        v1,v2,v3← coordinates of three vertices of face        $X_{\min}\leftarrow \mathrm{floor}\left[\min\left(v1_{X},v2_{X},v3_{X}\right)\right]$, $X_{\max}\leftarrow \mathrm{ciel}\left[\max\left(v1_{X},v2_{X},v3_{X}\right)\right]$        $Y_{\min}\leftarrow \mathrm{floor}\left[\min\left(v1_{Y},v2_{Y},v3_{Y}\right)\right]$, $Y_{\max}\leftarrow \mathrm{ciel}\left[\max\left(v1_{Y},v2_{Y},v3_{Y}\right)\right]$        $Z_{\min}\leftarrow \mathrm{floor}\left[\min\left(v1_{Z},v2_{Z},v3_{Z}\right)\right]$, $Z_{\max}\leftarrow \mathrm{ciel}\left[\max\left(v1_{Z},v2_{Z},v3_{Z}\right)\right]$        listOfBoxes= all boxes with index (p,q,r) with *p*$\in [X_{\min},X_{\max}]$, *q*$\in [Y_{\min},Y_{\max}]$, *r*$\in [Z_{\min},Z_{\max}]$.        Append {face:listOfBoxes} to the temporary dictionary data-structure faceList.    **end for**    **for** box in all boxes **do**        **for** *f* in faceList **do**            **if** box∈f.VALUES() **then**               Construct a hash table boxList with entries {box:f} or append *f* to box.VALUES() if entry with box key already exists.            **end if**        **end for**    **end for**    **Return**
 boxList

Once the hash table boxList is generated, the search algorithm that converts global to local coordinates consists of the following steps. We first use the global coordinates to determine the box in which the particle lies. This is a straightforward calculation as it entails using integer arithmetic. Then, the identified box is passed to the hash function, which returns the list of triangles associated with it. Finally, the ray intersection algorithm is used to search through the list of elements in the identified box. This final search returns the index of the triangle which contains the particle and the particle barycentric coordinates in the identified element. The overall search strategy is schematically shown in Figure 3.

The choice of the number of boxes (determined by $M_{X}$, $M_{Y}$, and $M_{Z}$) will determine the algorithm’s efficiency. At one extreme, $M_{X}=M_{Y}=M_{Z}=1$ (i.e., the mesh is contained in a single box) corresponds to a hash table with only one box containing all the faces. In this case, determining the particle barycentric coordinates and element would be equivalent to a brute-force search. The other extreme, consists in choosing large values for $M_{X}$, $M_{Y}$, and $M_{Z}$, so that few elements are contained in each box. In this case, the hash table will contain a large number of entries with the same elements contained in many boxes. This is also inefficient. Although $M_{X}$, $M_{Y}$, and $M_{Z}$ should be chosen depending on the number of elements used to discretize the domain of interest, in our numerical examples we have noticed that the efficiency of the search algorithm is not highly sensitive to these parameters. For most of the results presented in Section 3, we use a finite element mesh with 2562 nodes and 5120 triangular elements, and we adopt $M_{X}=M_{Y}=M_{Z}=15$. The resulting hash table (which does not include empty boxes) contained 1111 entries, each entry containing an average of 16 elements.

We conclude this section by noting that, due to the Lagrangian particle formulation, the particles’ global positions in the reference configuration are the input to the search algorithm. As a result, the hash table generated using Algorithm 1 needs to be *computed only once* for a given discretization of the reference sphere. Even as model parameters are modified, which, in turn, changes the substrate shape and the organization of the particles, the hash table does not need to be updated. Thus, the Lagrangian particle approach combined with this search algorithm offers a significant reduction in computational effort.

## 3. Results and Discussion

We now present computational results generated using the finite element formulation described above. We obtain these results by minimizing the discretized form of (Equation 26) using the weak form (Equation 27) with a gradient-based L-BFGS minimization solver [44]. A random initial state was used to initiate minimization. Unless otherwise stated, for all the results presented here, we used an icosahedrally symmetric spherical triangular mesh with 2562 nodes and 5120 elements. The chosen penalty parameters (cf., (Equation 26)) were $\mu_{1}=1000,\mu_{2}=1000$ and the preferred curvature $C_{0}$ was set equal to zero. Unless indicated otherwise, no volume constraint was imposed and the internal pressure was set to zero. Since the system’s topology was not allowed to change, the value of $\kappa_{g}$ does not influence the equilibrium state, a fact attributed to the Gauss–Bonnet theorem. Recall that the parameter $\lambda_{g}$ weighs the effect of the harmonic map energy and since this parameter appears as a penalty in Equation (Equation 26), a larger value is ideal but can cause slow convergence. In the following simulations, the value $\lambda_{g}$ was chosen to aid convergence. We remark that, once computed, the equilibrium configuration is not affected by the value $\lambda_{g}$.

Before presenting our results we note that caution should be exercised while visualizing the substrate/particle system, especially when coarser meshes are used. As shown in Figure 4a, when 642 nodes (and 1280 elements) were used to simulate a system with N=3 particles, the particles seem to lose contact with the surface. That is, the particles appear to lie inside the triangular enclosure, hovering below the surface, despite the Lagrangian particle framework being designed to prevent such situations. The cause of this apparent discrepancy stems from the visualization scheme used. In Figure 4a, only the nodal positions are used to generate the surface. However, the actual surface used by the Loop subdivision finite element scheme is the surface generated using box splines [34]. When this fact is taken into account, the discrepancy is resolved (see Figure 4b).

To validate the proposed method, we computed equilibrium configurations of particles interacting via pair-wise Coulombic interactions, i.e., Φ(r)=1/r. The results of our simulations are presented in Figure 5. For these simulations, we chose κ=1 and $\lambda_{g}=10$. We observe that the system replicates the symmetry configurations of the classical Thomson problem [46], which would correspond to the large bending stiffness limit. In the figure, we have employed the Schönflies notation for the symmetry groups: $D_{nh}$ represents an *n*-fold dihedral symmetry with a horizontal mirror plane, $O_{h}$ represents octahedral symmetry, etc.

Figure 6 shows the effect of changing the bending stiffness κ for N=12 particles interacting via a Lennard–Jones potential $\Phi \left(r\right)=\epsilon [{\left(r_{e}/r\right)}^{12}-2{\left(r_{e}/r\right)}^{6}]$. In these simulations ϵ=0.1, $\lambda_{g}=2$, $r_{e}=1.3$, and the particles are initially arranged in an icosahedrally symmetric configuration. κ was initially set equal to 3 and was gradually decreased while a numerical continuation scheme used the equilibrium configuration computed at each step as the initial guess for the subsequent step. As the bending stiffness κ decreases, a “stellated” particles/substrate configuration emerges, demonstrating the ability of the proposed formulation to model large shape changes and deformation of the coupled system.

The proposed formulation could be employed to study pinching of the substrate due to the particle interactions. We demonstrate this ability by analyzing a system with N=40 particles interacting via a harmonic interaction potential $V={\left(r-r_{e}\right)}^{2}/2$, with $r_{e}=0.8$. The other simulations parameters are κ=1, $\lambda_{g}=5$, and a volume constraint is also included. The state shown on the left of Figure 7 corresponds to a reduced volume v=0.95 (cf., Equation (Equation 5)), while the state on the right corresponds to a reduced volume v=0.90. These pinched states are reminiscent of viral budding or exo-cytosis. We note that it would not be possible to model the pinched neck state (shown in Figure 7, right) using the radial graph ansatz presented previously in [24] as the radial vector will no longer be a well-defined, single-valued function.

We conclude by presenting an example including a larger number of particles (N=200) interacting via the Lennard–Jones potential described above. Initially, particles are randomly distributed on a half of the reference sphere. Subsequently, their equilibrium distance $r_{e}$ is gradually increased from $r_{e}=0.15$ to $r_{e}=0.31$ in steps $\Delta r_{e}=0.01$, while the remaining model parameters remain constant (κ=2, ϵ=0.1, and $\lambda_{g}=10$). As $r_{e}$ increases, the particles move apart and start enveloping the substrate, until the full substrate is covered and deformed (Figure 8). This final example shows, once more, the ability of the proposed formulation to model the particles/substrate coupling and the effect of the particle arrangement on the substrate’s curvature.

## 4. Conclusions and Future Applications

This work presents a thin-shell finite element formulation combined with a Lagrangian particle framework to compute equilibrium states of interacting particles moving on a deformable substrate. We present the implementation of the formulation in the context of fluid shells, which we model with the Helfrich–Canham energy. Fluid shells are intrinsically degenerate structures, as any tangential motion along the surface of the shell does not contribute to its elastic energy. As a result, spurious zero-energy modes are present and commonly hinder the simulations of these systems. In this work, we adopt the *gauge-fixing* procedure recently developed in [27] to resolve the computational problems stemming from this degeneracy.

Since the particle positions are parametrized in the reference configuration, an attractive feature of the proposed formulation is that computation of terms in Equations (Equation 26) and (Equation 27) only requires the shape functions at locations on the reference mesh. An efficient search algorithm has been implemented to locate particles on the reference mesh by binning the mesh elements into boxes. Because of the referential description, the hash table generated for this purpose only needs to be created once.

This work generalizes and overcomes limitations of previous approaches to study particles moving of deformable substrates. Importantly, particles are not artificially pinned to the nodes of the substrate’s mesh, lifting a spurious constraint that limited the configurations achievable by the particles/substrate system. Furthermore, a radial graph ansatz was not necessary due to the gauge-fixing formulation that we have employed.

In this work, we disregard the dynamics of the system and solely focus on equilibrium configurations. Moreover, by employing the Helfrich model for the membrane, the present work focuses on idealized fluid membranes where in-plane viscosity is zero. However, it has been known that lipid membranes are viscoelastic [47,48]. In this regard, it is possible to extend the approach presented here to incorporate viscoelastic effects. Time dependence of the deformation map of the membrane must be assumed, i.e., f(X,t). Since the particle positions are parametrized in the reference configuration (see Section 2.2), they must also explicitly depend on time, i.e., $X_{i}\left(t\right)$. The absolute velocity of the particle is thus given by
(28)$${\dot{x}}_{i}\left(t\right)=\frac{d}{dt}f\left(X_{i}\left(t\right),t\right)=\nabla f\left(X_{i}\left(t\right),t\right){\dot{X}}_{i}\left(t\right)+\frac{\partial }{\partial t}f\left(X_{i}\left(t\right),t\right)\;,$$
where the dot represents the time derivative. Note that the second term in Equation (Equation 28) represents the velocity at the material point $X_{i}$ on the membrane, thus the relative velocity of the particle with respect to the membrane is given by the first term of Equation (Equation 28), i.e., $\nabla f\left(X_{i}\left(t\right),t\right){\dot{X}}_{i}\left(t\right)$. To model viscous drag on the particle due to the membrane, suitable models that depend on the relative velocity can be used. Recall that the gauge-fixing procedure had to be employed to circumvent problems due to the in-plane fluidity of the membrane. A viscoelastic model for the membrane obviates the need for this procedure.

Even though the finite element formulation has been presented for topologically spherical shells, the Lagrangian particle formulation and the search algorithm can be easily adapted to other topologies. The restriction to spherical surfaces is warranted by the gauge-fixing procedure that is employed to deal with fluid shells. As discussed in [27], this procedure is guaranteed to work only for spherical surfaces. However, the formulation presented above can be easily extended to other topologies in the case of solid shells, where material laws that include an in-plane stretching energy would prevent the tangential zero-energy modes typical of fluid shells.

Although in this work we only consider “zero temperature” equilibrium states, where thermal fluctuations are not accounted, the ultimate application of the presented formulation is to study the finite-temperature effects. Indeed, for many biophysical and soft-matter systems at physiologically relevant temperatures, thermal fluctuations play a key role [49,50]. The method presented here—where degrees of freedom are the nodal displacements of the mesh and the particles’ positions in the reference configuration—can be easily combined with standard Monte Carlo schemes used to simulate finite temperature systems. As the particles’ Monte Carlo moves occur in the reference configuration, locating the elements to which the particles belong is, once again, computationally inexpensive and based on the hash table generated only once for this purpose.

## Figures and Tables

**Figure 1 membranes-12-00960-f001:**
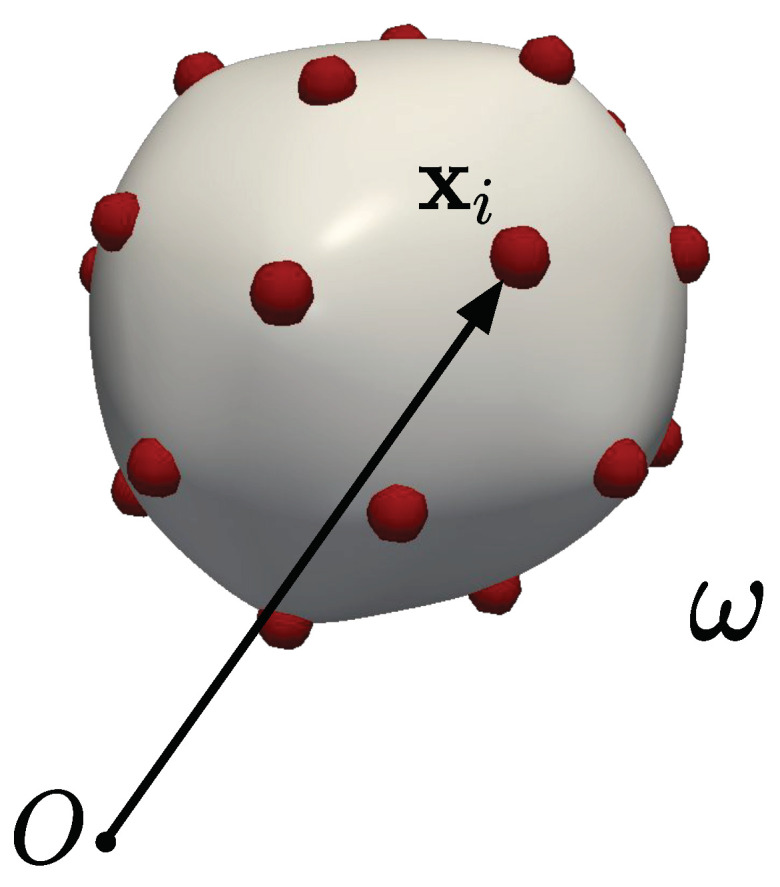
Schematic showing the deformed configuration of the membrane embedded with particles. Particle *i* has position vector $x_{i}$.

**Figure 2 membranes-12-00960-f002:**
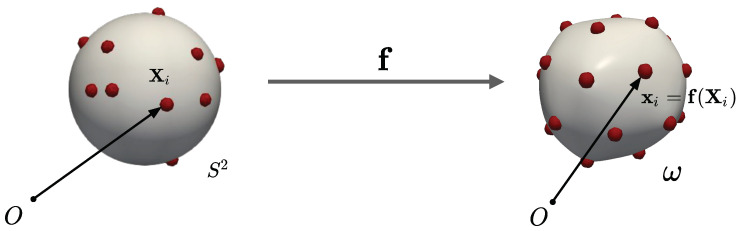
Reference and deformed configurations. The formulation degrees of freedom are the deformation mapping f(X) and the particle positions $X_{i}$ (i∈{1,2,⋯,N}) in the reference configuration.

**Figure 3 membranes-12-00960-f003:**
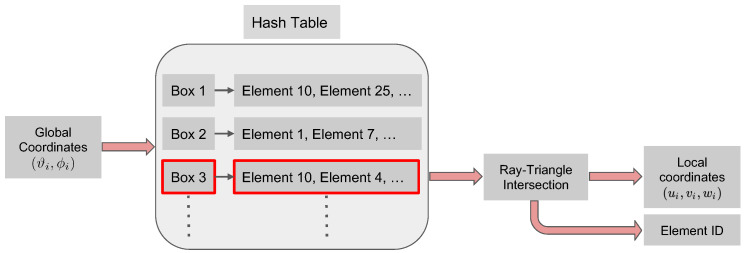
Schematic of the search algorithm used to convert the global coordinates $\left(\vartheta_{i},\phi_{i}\right)$ of particle *i* to its local barycentric coordinates $\left(u_{i},v_{i},w_{i}\right)$. The algorithm first locates the list of faces/element IDs (schematically highlighted in red) in the hash table (generated by Algorithm 1) and then searches this list with the ray-triangle intersection algorithm to compute the barycentric coordinates and finds the element ID.

**Figure 4 membranes-12-00960-f004:**
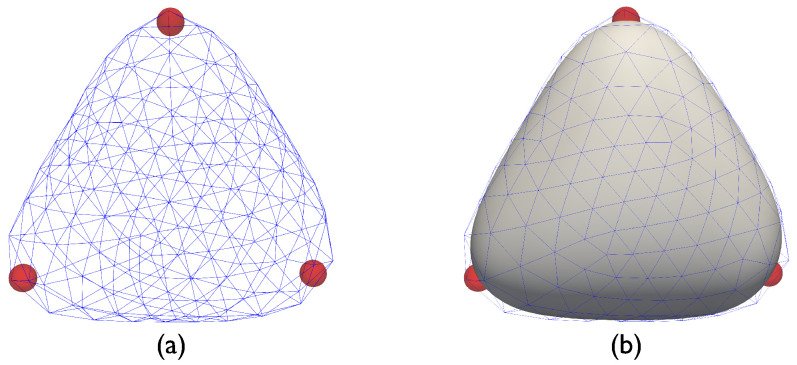
(**a**) Apparent loss of contact between particle and surface while visualizing the system using the underlying triangular mesh. (**b**) Actual surface shape after interpolation using Loop subdivision box splines (the underlying triangular mesh, shown in blue, is reported for reference).

**Figure 5 membranes-12-00960-f005:**
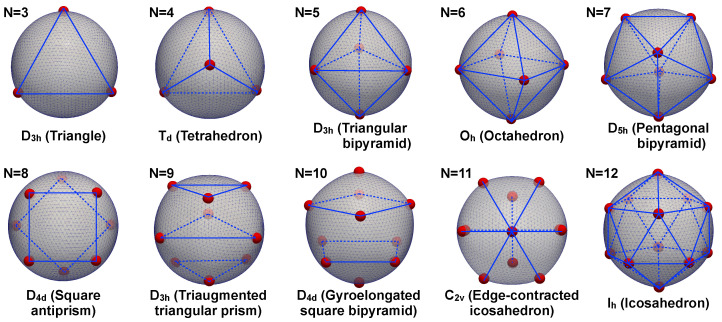
N particles interacting via Coulomb electric potential over a spherical surface with *high bending stiffness*. This case study approaches the classic Thomson problem where electric charges interact on a rigid sphere. The same symmetries computed in the Thomson problem are obtained for N=3 to N=12 [46], providing validation of the proposed approach. The finite element mesh used to discretize the underlying deformable substrate is shown together with lines (blue) to highlight the symmetry state of the computed particles (red) positions.

**Figure 6 membranes-12-00960-f006:**
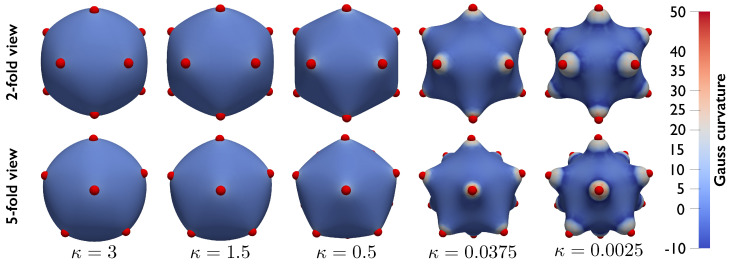
Particle−substrate configurations obtained while gradually decreasing the bending stiffness κ. The system is composed of N=12 particles interacting via Lennard–Jones potential and originally arranged in an icosahedral configuration. As the substrate bending stiffness decreases, the system configuration evolves from spherical to “stellated”, demonstrating that the proposed formulation can be employed to study large changes in the deformation and configurations of the particle/substrate system.

**Figure 7 membranes-12-00960-f007:**
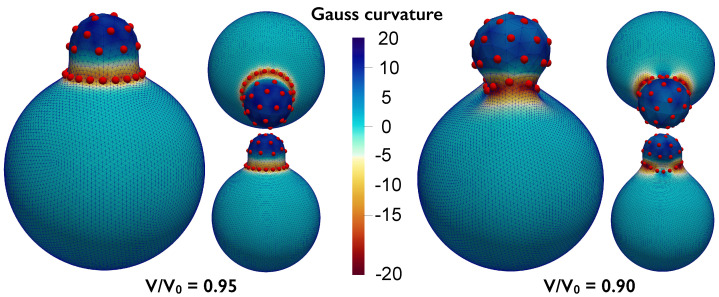
Particles interacting via harmonic potential over a deformable substrate with decreasing volume constraint. The volume is first decreased to 0.95 (**left**) and then to 0.9 (**right**) of the reference volume of a unit sphere (4/3π) producing budding of the underlying substrate. Lateral views from different angles (large panel and small bottom panel) and top/inclined view (small top panel) are provided. The mesh shown has been subdivided once according to the Loop scheme.

**Figure 8 membranes-12-00960-f008:**
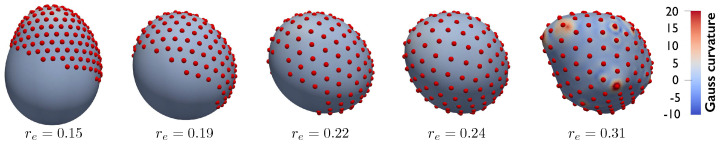
N=200 particles interacting via Lennard–Jones potential and gradually covering a deformable substrate. In the system’s initial configuration, the particles are present on a half of the reference sphere. The particles envelope and deform the full substrate as $r_{e}$ increases showing the strong particle−substrate coupling.

## Data Availability

Not applicable.

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
