# Peer review of "A Lagrangian Thin-Shell Finite Element Method for Interacting Particles on Fluid Membranes"

_membranes, 2022, doi:10.3390/membranes12100960_

Round 1

Reviewer 1 Report

This work presents a thin-shell finite element formulation combined with a Lagrangian particle framework to compute the equilibrium configurations of interacting particles moving on a deformable membrane. I think this overall is an interesting work. The authors did a good job in introducting the background of the problem concerned as well as explaining essential ingriendetns of the method. Below are my suggestions for the authors to consider.

1.       Proper references should be provided in the main texts from line 18 to 23 on page 1.

2.    It is better to move Fig. 2 right after Eq. (5) and then explain how this equation was derived. For example, what does “unit normal field” mean? Also, where does the factor 1/3 on the left hand side of Eq. (5) come from?

3.    In section 2, the authors "assume that the particles do not experience in-plane viscous forces". In reality, it is commonly believed that the diffusion of transmembrane proteins is critical for integrin clustering (PLoS Computational Biology, 2009, 5: e1000604) or aggregation of molecular bonds (Soft Matter. 2015, 11: 2812-2820; Soft Matter. 2016, 12: 4527-4583) often observed in cell adhesion. Can the present method be extended/modified to study these important phenomena?

4.    Given the extreme low bending rigidity of lipid cell membrane, thermal excitation alone could considerably deform the membrane (J. Mech. Phys. Solids, 2008, 56: 241-250; Physical Review Letters, 1994, 72:168–171) when then will likely affect the distribution of particles on the membrane. The authors may want to discuss these scenarios and give readers a more complete picture.

Reviewer 2 Report

This paper presents a thin-shell finite element formulation to calculate equilibrium configurations of a thin fluid shell with consideration of embedded particles. The paper is well organized. I think it can be published after a minor revision.

Comments

1. A Lagrangian formulation for interacting particles on a deformable medium is proposed in [1]. Moreover, a gauge-fixing procedure for spherical fluid membranes and application to computations is developed in [2]. Is it based on finite element method? What are the difference and innovation in comparison to the published work? The clear explanation is necessary to show the importance and the purpose of the current work.

[1] Dharmavaram, S.; Perotti, L.E. A Lagrangian formulation for interacting particles on a deformable medium. Computer Methods in Applied Mechanics and Engineering 2020, 364, 112949.

[2] Dharmavaram, S. A gauge-fixing procedure for spherical fluid membranes and application to computations. Computer Methods in 509 Applied Mechanics and Engineering 2021, 381, 113849.

2. The authors use many words to explain that adding EHM (eq. 7) to the free energy will not modify the surface constitutive law. Generally, equations are clearer to show some statements. So equations seem more suitable to demonstrate that the addition to the free energy will not modify the surface constitutive law.

3. The presented approach is not verified in the paper. It is better to give some verification.
